# Care of peripheral intravenous catheters in three hospitals in Spain: Mapping clinical outcomes and implementation of clinical practice guidelines

Ian Blanco-Mavillard[1,2,3]*, Gaizka Parra-García[4], Ismael Fernández-Fernández[1], Miguel Ángel Rodríguez-Calero[2,3,5], Celia Personat-Labrador[2], Enrique Castro-Sánchez[6]

1 Hospital Manacor, Manacor, Spain, 2 Universitat de les Illes Balears, Palma, Spain, 3 Care, Chronicity and Evidence in Health Research Group (CurES), Health Research Institute of the Balearic Islands (IdISBa), Palma, Spain, 4 Hospital Sant Joan de Deu, Palma, Spain, 5 Servei de Salut de les Illes Balears, Palma, Spain, 6 City University, London, United Kingdom

* ianblanco@hmanacor.org

## Abstract

### Background

Peripheral intravenous catheters (PIVCs) are the most widely used invasive devices worldwide. Up to 42% of PIVCs are prematurely removed during intravenous therapy due to failure. To date, there have been few systematic attempts in European hospitals to measure adherence to recommendations to mitigate PIVC failures.

### Aim

To analyse the clinical outcomes from clinical practice guideline recommendations for PIVC care on different hospital types and environments.

### Methods

We conducted an observational study in three hospitals in Spain from December 2017 to April 2018. The adherence to recommendations was monitored via visual inspection in situ evaluations of all PIVCs inserted in adults admitted. Context and clinical characteristics were collected by an evaluation tool, analysing data descriptively.

### Results

646 PIVCs inserted in 624 patients were monitored, which only 52.7% knew about their PIVC. Regarding PIVC insertion, 3.4% (22/646) patients had at least 2 PIVCs simultaneously. The majority of PIVCs were 20G (319/646; 49.4%) and were secured with transparent polyurethane dressing (605/646; 93.7%). Most PIVCs (357/646; 55.3%) had a free insertion site during the visual inspection at first sight. We identified 342/646 (53%) transparent dressings in optimal conditions (clean, dry, and intact dressing). PIVC dressings in medical wards were much more likely to be in intact conditions than those in surgical wards (234/

**Data Availability Statement:** All relevant data are within the paper and its Supporting Information files.

**Funding:** This work was supported by The College of Nurses of the Balearic Islands under grant number PI2017/0192. The findings and conclusions in this study are those of the authors and do not necessarily represent the official positions of The College of Nurses of the Balearic Islands. EC-S is also an NIHR Senior Nurse and Midwife Research Leader, and recognises the support of the NIHR Imperial Patient Safety Translational Research Centre and the BRC.

**Competing interests:** The authors declare that they have no competing interests.

**Abbreviations:** PIVC, Peripheral intravenous catheter; CRBSI, catheter-related bloodstream infection; CPG, Clinical practice guideline.

399, 58.7% vs. 108/247, 43.7%). We identified 55/646 (8.5%) PIVCs without infusion in the last 24 hours and 58/646 (9.0%) PIVCs without infusion for more than 24 hours. Regarding PIVC failure, 74 (11.5%) adverse events were identified, all of them reflecting clinical manifestation of phlebitis.

## Conclusions

Our findings indicate that the clinical outcome indicators from CPG for PIVC care were moderate, highlighting differences between hospital environments and types. Also, we observed that nearly 50% of patients did not know what a PIVC is.

## Background

Peripheral intravenous catheters (PIVCs) are among the most frequently used vascular access devices worldwide [1], with their insertion being one of the most common practices for hospital nurses. PIVCs are indicated for short-term use, usually around a week, for the administration of intravenous therapy [2]. It is estimated that physicians and nurses insert more than 330 million PIVCs annually in the United States [1, 3]. However, up to 42% of these catheters are prematurely removed during intravenous therapy due to PIVC failure [4], which is defined as the unplanned removal of the device due to mechanical complications (i.e., phlebitis, occlusion, infiltration) or infection before the completion of scheduled intravenous therapy [5, 6]. These complications are concerning in their own right, as catheter-related bloodstream infections (CRBSIs) are one of the most severe adverse events [7, 8], which can prolong hospital stay, resulting in attributable mortality rates of up to 25% [9, 10], and leading to unnecessary costs of approximately $45,000 per infection [5, 7].

In the last decade, healthcare systems have focused on reducing the variability of healthcare practice [11], implementing strategies to integrate the best recommendations of clinical practice guidelines (CPG) in combination with professional experience and user preferences [12–14], to provide optimally and quality care to patients [15]. However, introducing innovations such as the recommendations endorsed by CPGs onto routine clinical practice remains a complex and arduous process that is not exempt from difficulties [16]. For example, the number of CPG has grown significantly to the extent of being unmanageable [17]. Another critical challenge is the frequent tardiness in the implementation of these recommendations into clinical practice, probably fuelled by perceptions of clinical experience as the main element in decision-making [18]. Despite efforts to reduce the research-practice gap, some studies suggest that 30–40% of patients still do not receive healthcare based on the best available evidence [19–21], suggesting the difficulty of its implementation [22].

This gap is a complex and multifaceted phenomenon, which requires a deep understanding of decision-making [18]. The use of a knowledge mobilization model could counteract this situation, including strategies to promote fidelity to recommendations, audit and feedback of compliance and health literacy of vascular access, as crucial elements of a multimodal intervention [23]. Such gap is, therefore, a significant threat to patient safety and healthcare efficiency [24, 25]. To date, there have been few systematic attempts to measure the adherence to recommendations regarding optimal PIVC care and to mitigate PIVC failures in European hospitals. Therefore, the purpose of this study was to analyse the clinical outcome indicators from CPG for the insertion, maintenance, and management of PIVCs on different hospital types and environments.

## Methods

We used the Strengthening the Reporting of Observational Studies in Epidemiology (STROBE) statement for the reporting of observational studies to assist the reporting of our results.

### Study design and participants

We performed a prospective multicentre observational study, where data collectors directly observed the PIVC in situ. We conducted the study in all hospital wards of three hospitals in Mallorca (Spain). Hospital 1 and Hospital 2 are public-funded acute care hospitals and serve a population of 150.000 and 130.000 inhabitants. These hospitals have 224 and 165-bed hospitals respectively, for all clinical specialities except cardiac, thoracic, and neurosurgery. Hospital 3 is a long-term care hospital and has 197 hospital beds, of which 117 are intended for the treatment of chronic disease and palliative population.

We used the convenience sampling method and included all adult patients (18-years or older admitted to any hospital wards of the three hospitals), who have one or more PIVCs in situ on the day the researchers were present via unannounced. Emergency, critical care, paediatric, maternity, perioperative, and operating room areas were excluded in the analysis of the adherence to recommendations, as PIVCs in those areas are routinely maintained for less than 24 hours.

### Data collection

We collected data using a case report form that we had developed to analyse the clinical outcomes for the care of PIVC from clinical practice guidelines recommendations [26, 27]. Table 1 describes the recommendations and their clinical outcome indicators.

The case report form consisted of 20-items constructed in 5 sections to respond to the recommendations and validated by the Content Validity Index [28] for items (I-CVI) and scales (S-CVI). The items were rated on a 4-point relevance scale, considering 3 or 4 as relevant. The score results of I-CVI and S-CVI were 0.97 and 0.90, respectively, suggesting a very high content validity. S1 Table offers the results of the CVI on the 20-item case report form by six clinical experts.

The data collection was conducted from December 2017 to April 2018 by six external researchers to the hospital. All of them were intentionally selected for their expertise and training in the management of vascular access and had more than 5-years of employment as registered nurses. The external researchers received one-week of face-to-face training and a protocol for completing the case report form. Also, they completed a full working day evaluation with a mentor before starting the study. These standards homogenised the responses, minimising the potential bias during the study period. The researchers also collected information about context characteristics (gender, age years, education, and years employed by registered nurses). We considered these characteristics as variables that influencing the adherence to CPG recommendations.

### PIVC care and maintenance

Nurses inserted and maintained all PIVCs following the existing hospital policy, being like CPG recommendations. In summary, skin preparation before insertion was carried out with 2% chlorhexidine in 70% isopropyl alcohol. All PIVCs were Introcan Safety$^{TM}$ (non-winged) catheters (B. Braun), with a needle-free valve directly connected to 10cm of extension tubing ending in a three-way connector (Becton Dickinson). A transparent dressing with

**Table 1. Selection of the indicators for the care of peripheral intravenous catheters from clinical practice guideline recommendations.**

| Sections | Indicators | Clinical practice guideline recommendations |
|---|---|---|
| Catheter adequacy and insertion | 1. Intravenous cannula size, n (%): *16/18/20/22/24 gauge.* | Selection of the appropriate peripheral intravenous catheter insertion site, assessing risks for infection, against the risks of mechanical complications and patient comfort. |
| | 2. Insertion site, n (%): *dorsum of hand /wrist /forearm/ antecubital fossa/upper arm/foot.* | |
| | 3. Indwelling time, n (%): *less 48 hours/between 48–96 hours/more 96 h.* | Use of the upper extremity, preferably the forearm for peripheral intravenous catheter insertion unless medically contraindicated. |
| Catheter and catheter site care | 4. Dressing type, n (%): *Sterile transparent bordered semi-permeable polyurethane / Sterile gauze.* | Use of a sterile, transparent, semi-permeable polyurethane dressing to cover the intravascular insertion site. |
| | 5. Dressing integrity, n (%): *Poor / Perfect.* | Change of transparent, semi-permeable polyurethane dressings every 7 days, or sooner, if it is no longer intact or if moisture collects under the dressing. |
| | 6. Causes of poor integrity, n (%): *Not intact / Not clean / Not dry / Hematic residues.* | |
| | 7. Acknowledgement of PIVC, n (%): *Yes / No* | Patient education on treatment targets, administration, infusion, associated complications, care and management of the catheter. |
| Catheter removal and replacement strategies | 8. Visual inspection of insertion site, at first sight, n (%): *Not visible/ Visible.* | Inspection of the peripheral intravenous catheter insertion site at a minimum during each shift, recording the Visual Infusion Phlebitis score and/or infiltration score. |
| | 9. PIVC securement, n (%): *Not securement / Tubular mesh / Elastic bandage / Steri-strip / Medical tape.* | |
| | 10. Securement hinders the visualization of insertion site, n (%): *Yes / No.* | |
| | 11. Presence of adverse event during the visual inspection, n (%): *No / Persistent pain / Erythema and swelling / Palpable thrombosis / Deep venous thrombosis / Not defined.* | Surveillance for the occurrence of unexplained fever or pain at the insertion site, examining for the occurrence of redness, erythema, or inflammation. |
| | | Removal of the peripheral intravenous catheter when complications occur, or as soon as it is no longer required. |
| | 12. Infusion type, n (%): *Continuous infusion / Intermittent Infusion / In bolus / No infusion in less than 24 hours / No infusion for more than 24 h.* | Removal of the unnecessary peripheral intravenous catheter, when intravenous treatment is not administered after 24 h. |
| Record and documentation PIVC care | 13. Presence of PIVC insertion records, n (%): *Yes / No.* | Record of peripheral intravenous catheter insertion, including assessment of insertion site and functionality. |
| | 14. Dressing date recorded, n (%): *Yes / No.* | Documentation of peripheral intravenous catheter insertion or maintenance date at the transparent dressing. |

polyurethane borders (Tegaderm™, 3M) was applied at the insertion site to secure the PIVC in situ. Standard caps on all needleless connectors were in place to minimise accidental tubing disconnections.

## Ethical considerations

The research ethic committee of Hospital Manacor and Balearic Islands approved this study (IB3492/17PI). All patients were informed about the purpose of the study and their implications. We obtained oral consent from patients or their legal guardian (in the case of patients with cognitive impairment) before study participation. Patients who accepted to participate in the study were progressively included during the study period. No patient refused to participate in the study.

## Statistical analysis

The statistical analysis included a description of the sample (continuous data represented by means and standard deviation, and categorical data represented by frequency and percentage), and bivariate analysis with parametric and non-parametric tests, depending on the nature of the distributions (correlation, ANOVA, chi-square). Data were analysed using SPSS IBM Statistics version 25.

## Results

### Characteristics of the context

Thirteen hospital wards participated in this study, of which 5/13 (38.4%) from hospital 1 and 3, and 3/13 (23.1%) from hospital 2. Most wards were medical (9/13; 69.2%). One hundred fifty-eight nurses participated in the study, of whom 139 (88%) were female nurses between 31 to 40 years old and 11 to 20 years employed as a registered nurse. During the development of the study, we analysed 624 patients, again mainly from medical wards (393; 63%). There were 265/624 (42.5%) female patients, with a mean age of 71.0 years (SD, 14.8 years). In our sample, 474 patients (76%) did not present cognitive impairment. Among these, 250 patients (52.7%) recognized and identified the inserted PIVC. We observed significant differences between the characteristics of context, nurses' age (p < 0.001), professional experience (p < 0.001), patient age (p = 0.005), patient cognitive impairment (p < 0.001), and acknowledgement of inserted PIVC (p < 0.001) comparing hospital types. Also, there were statistically significant differences between nurses' age (p = 0.017), and acknowledgement of inserted PIVC (p < 0.001) comparing hospital environments. All variables associated with characteristics of context are described in Table 2.

**Table 2. Comparative analysis of characteristics of context.**

| Characteristics of context | Overall | Hospital types | | | | Hospital environments | | |
|---|---|---|---|---|---|---|---|---|
| | | Hospital 1 | Hospital 2 | Hospital 3 | p-value | Medical ward | Surgical ward | p-value |
| Total wards, n (%) | 13 (100) | 5 (38.4) | 3 (23.1) | 5 (38.4) | | 9 (69.2) | 4 (30.8) | |
| Total nurses, n (%) | 158 (100) | 60 (38.0) | 44 (27.8) | 54 (34.2) | | 124 (78.5) | 34 (21.5) | |
| Nurses gender, n (%) | | | | | 0.444 | | | 0.255 |
| Female | 139 (88.0) | 52 (86.8) | 41 (93.2) | 46 (85.2) | | 111 (89.5) | 28 (82.4) | |
| Male | 19 (12.0) | 8 (13.3) | 3 (6.8) | 8 (14.8) | | 13 (10.5) | 6 (17.6) | |
| Nurses age (years), n (%) | | | | | < 0.001 | | | 0.017 |
| 21–25 | 19 (12.0) | 5 (8.3) | 0 | 14 (26.0) | | 12 (9.7) | 7 (20.6) | |
| 26–30 | 46 (29.1) | 19 (31.7) | 6 (13.6) | 21 (38.9) | | 37 (29.8) | 9 (26.5) | |
| 31–40 | 77 (48.7) | 30 (50.0) | 32 (72.7) | 15 (27.8) | | 65 (52.4) | 12 (35.6) | |
| 41–50 | 14 (8.9) | 6 (10.0) | 4 (9.1) | 4 (7.4) | | 10 (8.1) | 4 (11.8) | |
| 50–60 | 2 (1.3) | 0 | 2 (4.5) | 0 | | 0 | 2 (5.9) | |
| Nurses academic level | | | | | 0.293 | | | 0.425 |
| Bachelor Nurse | 152 (96.2) | 56 (93.3) | 44 (100) | 52 (96.3) | | 118 (95.2) | 34 (100) | |
| Master of Science | 6 (3.8) | 4 (6.6) | 0 | 2 (3.7) | | 6 (4.8) | 0 | |
| Years employed as nurse | | | | | <0.001 | | | 0.787 |
| 0–5 | 42 (26.6) | 2 (3.3) | 4 (9.1) | 36 (66.7) | | 32 (25.8) | 10 (29.5) | |
| 6–10 | 31 (19.6) | 7 (11.7) | 11 (25.0) | 13 (24.1) | | 23 (18.5) | 8 (23.5) | |
| 11–20 | 61 (38.6) | 29 (48.3) | 27 (61.4) | 5 (9.2) | | 48 (38.7) | 13 (38.2) | |
| 21–30 | 24 (15.2) | 22 (36.7) | 2 (4.5) | 0 | | 21 (16.9) | 3 (8.8) | |
| Total patients, n (%) | 624 (100) | 277 (63.0) | 211 (37.0) | 158 (37.0) | | 393 (63.0) | 231 (37.0) | |
| Patient gender, n (%) | | | | | 0.340 | | | 0.750 |
| Female | 265 (42.5) | 110 (42.0) | 81 (39.3) | 74 (47.1) | | 165 (26.5) | 100 (16.0) | |
| Male | 359 (57.5) | 152 (58.0) | 124 (60.7) | 83 (52.9) | | 228 (36.5) | 131 (21.0) | |
| Patient age (years), mean (SD) | 71.0 (14.8) | 69.1 (15.4) | 71.7 (12.4) | 73.1 (16.1) | 0.005 | 72.6 (14.5) | 68.2 (14.7) | 0.184 |
| Cognitive impairment, n (%) | | | | | < 0.001 | | | 0.098 |
| Yes | 150 (24.0) | 40 (15.3) | 66 (32.2) | 44 (28.0) | | 103 (16.5) | 47 (7.5) | |
| No | 474 (76.0) | 222 (84.7) | 139 (67.8) | 113 (72.0) | | 290 (46.5) | 184 (29.5) | |
| Acknowledgement of PIVC | | | | | < 0.001 | | | <0.001 |
| No | 224 (47.3) | 108 (48.6) | 52 (37.4) | 64 (56.6) | | 161 (55.5) | 63 (34.2) | |
| Yes | 250 (52.7) | 114 (51.4) | 87 (62.6) | 49 (43.4) | | 129 (44.5) | 121 (65.8) | |

## Clinical characteristics and outcomes of the sample

During the development of the study, we analysed 646 PIVCs in situ from 624 patients, of which had at least 1 PIVC (96.6%), and 22 patients had 2 PIVCs (3.4%). A high number of PIVCs (274/646; 42.4%) were inserted in a non-flexure anatomical site, such as the forearm, inserted mostly from the hospital ward (347/646, 53.7%). In terms of catheter size, the majority were 20G (319/646, 49.4%). A high proportion of PIVCs (373/646, 57.7%) had been in situ for less than 48 hours at the time of the evaluation. There were statistically significant differences between the hospital environments regarding total PIVC in situ per patient ($p = 0.001$), insertion site ($p = 0.005$), cannula size ($p < 0.001$), indwelling time ($p < 0.001$) and setting of insertion ($p<0.001$). Also, there were statistically significant differences between the hospital types regarding total PIVC in situ per patient ($p = 0.026$), insertion site ($p < 0.001$), cannula size ($p < 0.001$), indwelling time ($p = 0.023$) and setting of insertion ($p<0.001$). All variables associated with clinical characteristics and outcomes are described in Table 3.

**Table 3. Clinical characteristics of the sample.**

| Clinical characteristics | Overall | Hospital types | | | | Hospital environments | | |
|---|---|---|---|---|---|---|---|---|
| | | Hospital 1 | Hospital 2 | Hospital 3 | p-value | Medical ward | Surgical ward | p-value |
| Total PIVCs, n (%) | 646 (100) | 277 (42.9) | 211 (32.7) | 158 (24.4) | | 399 (61.8) | 247 (38.2) | |
| Total PIVC in situ / patient | | | | | 0.026 | | | 0.001 |
| 1 PIVC | 624 (96.6) | 262 (94.6) | 205 (97.2) | 157 (99.4) | | 393 (98.5) | 231 (93.5) | |
| 2 PIVC | 22 (3.4) | 15 (5.4%) | 6 (2.8) | 1 (0.6) | | 6 (1.5) | 16 (6.5) | |
| Insertion site, n (%) | | | | | < 0.001 | | | 0.005 |
| Hand | 154 (23.8) | 64 (23.1) | 46 (21.8) | 44 (27.8) | | 82 (12.7) | 72 (11.1) | |
| Wrist | 83 (12.8) | 28 (10.1) | 21 (10.0) | 34 (21.5) | | 51 (7.9) | 32 (5.0) | |
| Forearm | 274 (42.4) | 114 (41.2) | 96 (45.5) | 64 (40.5) | | 186 (28.8) | 88 (13.6) | |
| Antecubital fossa | 114 (17.6) | 67 (24.2) | 42 (19.9) | 5 (3.2) | | 62 (9.6) | 52 (8.0) | |
| Arm | 16 (2.5) | 3 (1.1) | 6 (2.8) | 7 (4.4) | | 14 (2.2) | 2 (0.3) | |
| Foot | 5 (0.8) | 1 (0.4) | 0 | 4 (2.5) | | 4 (0.6) | 1 (0.2) | |
| IV cannula size, n (%) | | | | | < 0.001 | | | < 0.001 |
| 16 gauge | 3 (0.5) | 0 | 0 | 3 (1.9) | | 0 | 3 (0.5) | |
| 18 gauge | 137 (21.2) | 57 (20.6) | 43 (20.4) | 37 (23.4) | | 42 (6.5) | 95 (14.7) | |
| 20 gauge | 319 (49.4) | 146 (52.7) | 104 (49.3) | 69 (43.7) | | 223 (34.5) | 96 (14.9) | |
| 22 gauge | 94 (14.6) | 19 (6.9) | 31 (14.7) | 44 (27.8) | | 68 (10.5) | 26 (4.0) | |
| 24 gauge | 3 (0.5) | 1 (0.4) | 2 (0.9) | 0 | | 2 (0.3) | 1 (0.2) | |
| Unevaluated | 90 (13.9) | 54 (19.5) | 31 (14.7) | 5 (3.2) | | 64 (9.9) | 26 (4.0) | |
| Indwelling time | | | | | 0.023 | | | < 0.001 |
| < 48 hours | 373 (57.7) | 168 (60.6) | 105 (49.8) | 100 (63.3) | | 208 (52.1) | 165 (66.8) | |
| 48–96 hours | 138 (21.4) | 62 (22.4) | 49 (23.2) | 27 (17.1) | | 90 (22.6) | 48 (19.4) | |
| > 96 hours | 135 (20.9) | 47 (17.0) | 57 (27.0) | 31 (19.6) | | 101 (25.3) | 34 (13.8) | |
| Setting of Insertion, n (%) | | | | | < 0.001 | | | < 0.001 |
| Hospital Ward | 347 (53.7) | 120 (43.3) | 101 (47.9) | 127 (80.4) | | 261 (42.5) | 86 (13.3) | |
| Operating room | 80 (12.4) | 29 (10.5) | 23 (10.9) | 28 (17.7) | | 1 (0.2) | 79 (12.2) | |
| Emergency dep. | 177 (37.4) | 103 (37.2) | 74 (35.1) | 0 | | 111 (18.1) | 66 (10.2) | |
| Intensive Care unit | 2 (0.3) | 0 | 2 (0.9) | 0 | | 2 (0.3) | 0 | |
| Ambulatory unit | 2 (0.3) | 0 | 2 (0.9) | 0 | | 0 | 2 (0.3) | |
| Primary care | 6 (0.9) | 2 (0.7) | 2 (0.9) | 1 (0.6) | | 4 (0.7) | 2 (0.3) | |
| Not registered | 32 (5.0) | 23 (8.3) | 7 (3.3) | 2 (1.3) | | 20 (3.1) | 12 (1.9) | |

Table 4 offers all information of PIVC indicators from clinical guideline by hospital types and settings. The insertion site must be freely inspected at first sight and, at minimum, during each shift or administration of intravenous therapy for successful PIVC maintenance. These requirements were the case for most PIVCs (357/646; 55.3%). All visible PIVCs (605/646; 93.7%) had a transparent bordered polyurethane dressing while the rest (41/646; 6.3%) were not visible. Also, a higher number of PIVCs (231/646; 35.9%) had two or more types of securement, and 79/646 (12.2%) were entirely covered by an elastic bandage. There were statically significant differences between hospital types regarding visualization at first sight (p < 0.001) and PIVC securement preventing such visualization of the insertion site (p = 0.002). However, there were no statically significant differences between these same outcomes comparing the hospital environments.

**Table 4. Comparative analysis of PIVC indicators from clinical guideline by hospital types and settings.**

| Variables | Overall | Hospital types | | | | Hospital environments | | |
|---|---|---|---|---|---|---|---|---|
| | | Hospital 1 | Hospital 2 | Hospital 3 | p-value | Medical ward | Surgical ward | p-value |
| Total PIVCs, n (%) | 646 (100) | 277 (42.9) | 211 (32.7) | 158 (24.4) | | 399 (61.8) | 247 (38.2) | |
| Visual inspection of insertion site, at first sight, n (%) | | | | | < 0.001 | | | 0.935 |
| Not visible | 289 (44.7) | 143 (51.6) | 101 (47.9) | 45 (28.5) | | 179 (44.9) | 110 (44.5) | |
| Visible | 357 (55.3) | 134 (48.4) | 110 (52.1) | 113 (71.5) | | 220 (55.1) | 137 (55.5) | |
| Dressing type, n (%) | | | | | 0.200 | | | 0.004 |
| Transparent bordered polyurethane | 605 (93.7) | 254 (91.7) | 200 (94.8) | 151 (95.6) | | 363 (91.5) | 240 (97.2) | |
| Gauze | 0 | 0 | 0 | 0 | | 0 | 0 | |
| Not visible | 41 (6.3) | 23 (8.3) | 11 (5.2) | 7 (2.8) | | 34 (8.5) | 7 (2.8) | |
| Dressing integrity, n (%) | | | | | < 0.001 | | | < 0.001 |
| Poor | 222 (34.4) | 112 (40.4) | 60 (28.4) | 49 (31.0) | | 103 (25.8) | 118 (47.8) | |
| Perfect (clean, dry and intact) | 341 (52.8) | 122 (44.0) | 134 (63.5) | 86 (54.4) | | 234 (58.7) | 108 (43.7) | |
| Unevaluated | 83 (12.8) | 43 (15.5) | 17 (8.1) | 23 (14.6) | | 62 (15.5) | 21 (8.5) | |
| Causes of poor integrity, n (%) | | | | | 0.068 | | | < 0.001 |
| Not intact | 42 (19.0) | 22 (19.6) | 11 (18.3) | 9 (18.4) | | 11 (10.7) | 31 (26.3) | |
| Not clean | 11 (5.0) | 2 (1.8) | 6 (10.0) | 3 (6.1) | | 9 (8.7) | 2 (1.7) | |
| Not dry | 32 (14.5) | 11 (9.8) | 15 (25.0) | 6 (12.2) | | 21 (20.4) | 11 (9.3) | |
| Hematic residues | 76 (34.4) | 37 (33.0) | 18 (30.0) | 21 (42.9) | | 41 (39.8) | 35 (29.7) | |
| Two or more | 60 (27.1) | 40 (35.7) | 10 (16.7) | 10 (20.4) | | 21 (20.4) | 39 (33.0) | |
| PIVC securement, n (%) | | | | | < 0.001 | | | < 0.001 |
| Not securement | 70 (10.8) | 19 (6.9) | 21 (10.0) | 30 (19.0) | | 44 (11.0) | 26 (10.5) | |
| Tubular mesh | 125 (19.3) | 52 (18.8) | 60 (28.4) | 13 (8.2) | | 120 (30.1) | 5 (2.0) | |
| Elastic bandage | 79 (12.2) | 33 (11.9) | 19 (9.0) | 27 (17.1) | | 66 (16.5) | 13 (5.3) | |
| Steri-strip | 80 (12.4) | 25 (9.0) | 26 (12.3) | 29 (18.4) | | 28 (7.0) | 52 (21.1) | |
| Medical tape | 61 (9.4) | 5 (1.8) | 43 (20.4) | 13 (8.2) | | 47 (11.8) | 14 (5.7) | |
| Two or more | 231 (35.9) | 143 (51.6) | 42 (19.9) | 46 (29.1) | | 94 (23.6) | 137 (55.4) | |
| PIVC securement hinders the visualization of insertion site, n (%) | | | | | 0.002 | | | 0.338 |

*(Continued)*

**Table 4.** (Continued)

| Variables | Overall | Hospital types | | | | Hospital environments | | |
| --- | --- | --- | --- | --- | --- | --- | --- | --- |
| | | Hospital 1 | Hospital 2 | Hospital 3 | p-value | Medical ward | Surgical ward | p-value |
| No | 415 (64.2) | 159 (57.4) | 139 (65.9) | 117 (74.1) | | 262 (65.7) | 153 (61.9) | |
| Yes | 231 (35.8) | 118 (42.6) | 72 (34.1) | 41 (25.9) | | 137 (34.3) | 94 (38.1) | |
| Infusion type, n (%) | | | | | < 0.001 | | | 0.141 |
| Continuous infusion | 233 (36.1) | 80 (28.9) | 76 (36.0) | 77 (48.7) | | 141 (35.3) | 92 (37.2) | |
| Intermittent Infusion | 263 (40.7) | 127 (45.8) | 76 (36.0) | 60 (38.0) | | 171 (42.9) | 92 (37.2) | |
| In bolus | 37 (5.7) | 18 (6.5) | 18 (8.5) | 1 (0.6) | | 27 (6.8) | 10 (4.0) | |
| No infusion in less than 24 h | 55 (8.5) | 16 (5.8) | 23 (10.9) | 16 (10.1) | | 29 (4.5) | 26 (10.5) | |
| No infusion for more than 24 h | 58 (9.0) | 36 (13.0) | 18 (8.5) | 4 (2.5) | | 31 (7.8) | 27 (10.9) | |
| PIVC failure, n (%) | | | | | 0.031 | | | 0.086 |
| No | 364 (56.3) | 142 (51.3) | 117 (55.4) | 105 (66.4) | | 237 (59.4) | 127 (51.4) | |
| Yes | 74 (11.5) | 32 (11.5) | 28 (13.3) | 14 (8.9) | | 46 (11.5) | 28 (11.3) | |
| Unevaluated | 208 (32.2) | 103 (37.2) | 66 (31.3) | 39 (24.7) | | 116 (29.1) | 92 (37.1) | |
| Presence of adverse event during visual inspection, n (%) | | | | | 0.019 | | | 0.066 |
| Persistent pain | 29 (4.5) | 10 (3.6) | 13 (6.2) | 6 (3.8) | | 15 (3.8) | 14 (5.7) | |
| Erythema | 41 (6.3) | 22 (7.9) | 11 (5.2) | 8 (5.1) | | 28 (7.0) | 13 (5.3) | |
| Palpable thrombosis | 3 (0.5) | 0 | 3 (1.4) | 0 | | 3 (0.7) | 0 | |
| Deep venous thrombosis | 0 | 0 | 0 | 0 | | 0 | 0 | |
| Not defined | 1 (0.2) | 0 | 1 (0.5) | 0 | | 0 | 1 (0.4) | |
| Presence of PIVC insertion records, n (%) | | | | | < 0.001 | | | < 0.001 |
| No | 338 (52.3) | 220 (79.4) | 105 (49.8) | 13 (8.2) | | 170 (42.6) | 168 (68.0) | |
| Yes | 308 (47.7) | 57 (20.6) | 106 (50.2) | 45 (91.8) | | 229 (57.4) | 79 (32.0) | |
| Dressing date recorded, n (%) | | | | | < 0.001 | | | < 0.001 |
| No | 564 (87.3) | 264 (95.3) | 197 (93.4) | 103 (65.2) | | 320 (80.2) | 244 (98.8) | |
| Yes | 82 (12.7) | 13 (4.7) | 14 (6.6) | 55 (34.8) | | 79 (19.8) | 3 (1.2) | |

Regarding the clinical outcome indicators from CPG for the maintenance recommendations, we identified 341/646 (52.8%) transparent dressings in perfect conditions (defined as a clean, dry, and intact dressing). PIVC dressings in medical wards were much more likely to be in perfect conditions than those in surgical wards (234/399; 58.7% vs. 108/247; 43.7%). The most frequent defect seen in the dressings was the presence of blood traces or residues (76/221; 34.4%) inside the transparent dressing membrane or the combination of two or more conditions, such as not intact, not dry, or not clean (60/221; 27.1%). The variables related to dressing type (p = 0.004), integrity (p < 0.001), causes of poor integrity (p < 0.001) and PIVC securement (p < 0.001) were statistically significant depending on the hospital environments. However, there were no statically significant differences between dressing type (p = 0.200) and causes of poor integrity (p = 0.068) comparing the hospital types.

The most frequent type of intravenous infusion was intermittent (263/646, 40.7%). We identified 55/646 (8.5%) PIVCs without infusion in the last 24 hours and 58/646 (9.0%) PIVCs without infusion for more than 24 hours. There were statistically significant differences

between the hospital types regarding the infusion type (p < 0.001). However, there were no statically significant differences between this same outcome (p = 0.141) comparing the hospital environments.

Concerning PIVC failure, 74 (11.5%) all adverse events identified were clinical of manifestations of phlebitis. There were 29 episodes (4.5%) of persistent pain, 41 (6.3%) of erythema and swelling around the insertion site, and 3 (0.5%) thrombosis. However, 208 PIVCs (32.2%) could not be evaluated due to the presence of occlusive bandage. There were statically significant differences between hospital types regarding PIVC failure (p = 0.031) and the occurrence of adverse events (p = 0.019).

Less than 50% of nurses documented all information about PIVC insertion on the patient's clinical history (308/646, 47.7%). As for the recording of dressing dates, only 82/646 (12.7%) were documented on the transparent dressings. We observed a higher rate of PIVC insertion and dressing date records in medical wards (229/399; 57.4% and 79/399; 19.8% respectively) than in surgical wards (79/247; 32% and 3/247; 1.2% respectively). We observed a statistically significant association between hospital types and environments comparing the presence of PIVC insertion recording (p < 0.001) and dressing date (p < 0.001).

## Discussion

Our study focused on analysing the indicators from international CPG recommendations for the insertion, management, and care of PIVC comparing hospital types and environments. This study allowed us to map the baseline of the clinical outcomes for the implementation of these recommendations into clinical practice. The findings related to dressing status, visual inspection, and unnecessary PIVCs were moderate. International evidence recommends that PIVC dressings should be intact, clean, and dry, plus adequately secured and visible during the inspection of insertion site for prevention of PIVC failure [29–31]. The 34% of transparent dressings were not in optimal conditions, a rate comparable to the 21–34% of dressings compromised (moist, soiled, inadequately secured, or lifting of the skin) reported previously [1, 32]. An optimum dressing should reduce multiple complications of PIVCs, such as extravasation or dislodgement due to micromotions of the PIVC within the vein. Also, the poor dressing status may cause the colonization of microorganisms, such as S. aureus, which can lead to severe complications and death [33]. In our study, we observed that approximately 45% of PIVC insertion sites were not visible, a disappointing scenario considering the impact that inspection of the PIVC insertion site per shift would have to prevent and mitigate adverse events [31]. Removal of the PIVC should occur if phlebitis, inflammation or obstruction are present, or intravenous therapy has completed in the previous 24 hours, or the PIVC is no longer needed [4, 23, 34]. However, 9% of total PIVCs were unnecessarily maintained, with no differences between these outcomes and the hospital environment. These results provide insight into the various mechanisms that lead to PIVC failure, reflecting a poor adherence to recommendations with associated iatrogenic harms to patients [35]. PIVC failure triggers the need for a new PIVC insertion with its potential adverse events, increasing the risk of CRBSIs [4], as well as imposing a significant demand on healthcare resources [36, 37]. We observed suboptimal performance between different hospital environments and types regarding the clinical outcomes from CPG recommendations for care of PIVC. However, information regarding the use and knowledge of evidence, the level of burnout or dissatisfaction, the level of workload within the unit, and/or culture of implementation would be required to determine the conditions underlying this suboptimal performance concerning different environments [38].

From the point of view of safety and quality of care, healthcare organizations should emphasize clinical implementation strategies [39] and delve deeper into their understanding

of internal decision-making processes [40]. This strategy should incorporate mechanisms to mediate knowledge into decision-making [41, 42], which is not only achieved through the careful selection of evidence but also the weight of multiple humans factors [22]. Scientific evidence should receive significant attention, but optimal decisions would require the integration of such evidence with clinical experience together with patient involvement in shared decision-making [12, 43], shaping and coproducing practice "mindlines" [18, 41, 44, 45]. However, we observed that almost 47.3% of patients did not know what a PIVC was. This finding reflects a wide gap in patients' knowledge regarding PIVC used in their care. Health literacy can be a driver of change to empower patients in this self-care, being the initial hurdle to prevent and monitor the development of complications and adverse events [46, 47]. Therefore, health literacy of vascular access is a fundamental element that should be included within multimodal interventions to improve catheter failure outcomes, patient empowerment in their self-care, and shared decision making. Research efforts are needed to conduct this future education as an improvement for patient and self-care of vascular access.

This study was essential to assess the clinical outcome indicators from CPG recommendations regarding the hospital environment as a baseline within a multimodal intervention [48]. We should integrate the best evidence while deepening the motivations and beliefs of health professionals during decision-making for successful implementation, but we must also pay attention to the context, the great neglected in the field of implementation science [49]. The lack of context-related knowledge may be one of the most significant problems for implementation strategies into the healthcare system, being itself a fundamental limitation that questions the efforts made to improve the quality of interventions and the fidelity on the use of evidence-based practice [50]. The adaptation of interventions to local contexts is an indispensable element in the science of implementation. However, it is frequently not explicitly considered how local context factors determine its success [49]. In-depth knowledge of nurse participation in hospital affairs, leadership, size of nursing teams and professional relationship, added to the use, attitudes, and knowledge of professionals towards evidence-based practice will provide relevant information about the contextual and individual mechanisms [51, 52] that can facilitate the integration of tacit and explicit knowledge into decision-making improving adherence to best available recommendations [49, 53].

Our study presents some limitations. Our method could not include the evaluation of some relevant practices that could influence the appearance of CRBSIs and PIVC failures, such as the care of the patient's catheter hub and connection port and using flush solutions to maintain the permeability of PIVCs during evaluation. Future research must consider analysis and integration of contextual and individual factors on the use of best available knowledge in clinical practice decisions to improve adherence to insertion and management recommendations of PIVCs as a critical element to be considered within multimodal strategies for effective knowledge mobilization [54, 55]. Also, these quality improvement initiatives for PIVC care should include a set of relevant interventions consisting of the adequacy of the vascular access device, optimal PIVC insertion care, maintenance and management of intravenous therapy, proactive pursuit of opportunities for removing unnecessary PIVCs and health literacy of vascular access. These actions would improve PIVC failure outcomes, sharing decision making, and lead to significant cost savings for healthcare systems.

## Conclusions

Our findings indicate that the clinical outcome indicators from CPG for the insertion, management, and maintenance associated with PIVC were moderate, highlighting differences between hospital environments and types. Also, we observed that almost 50% of patients did

not know what a PIVC is. These findings reflect a wide gap between knowledge and optimal clinical practice, which would explain the moderate adherence to PIVC care and the need to use a knowledge transfer model with contextual mechanisms and individual factors.

## Supporting information

**S1 Table. Ratings on a 20-item case report form by clinical experts.**
(PDF)

## Acknowledgments

We sincerely thank Ms. Francesca Rosa Rosal-Obrador, Dr. Concepcion Zaforteza-Lallemand, Ms. Isabel Roman Medina, and Dr. Joan Ernest de Pedro-Gómez (Director of Nursing of Hospital Manacor, Hospital Comarcal de Inca and Hospital Sant Joan de Deu, and Lecturer of Universitat de les Illes Balears respectively) for their support. We would also like to sincerely thank all external researchers and patients of three hospitals for their crucial assistance in developing this study.

## Author Contributions

**Conceptualization:** Ian Blanco-Mavillard, Miguel Ángel Rodríguez-Calero, Enrique Castro-Sánchez.

**Formal analysis:** Ian Blanco-Mavillard, Ismael Fernández-Fernández, Enrique Castro-Sánchez.

**Funding acquisition:** Miguel Ángel Rodríguez-Calero, Enrique Castro-Sánchez.

**Investigation:** Ian Blanco-Mavillard.

**Methodology:** Ian Blanco-Mavillard, Ismael Fernández-Fernández, Miguel Ángel Rodríguez-Calero, Enrique Castro-Sánchez.

**Supervision:** Ian Blanco-Mavillard, Gaizka Parra-García.

**Writing – original draft:** Ian Blanco-Mavillard.

**Writing – review & editing:** Ian Blanco-Mavillard, Gaizka Parra-García, Ismael Fernández-Fernández, Miguel Ángel Rodríguez-Calero, Celia Personat-Labrador, Enrique Castro-Sánchez.

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
