## [Decision Letter · Decision Letter 0]

3 Aug 2020

PONE-D-20-15991

CARE OF PERIPHERAL INTRAVENOUS CATHETERS IN THREE HOSPITALS IN SPAIN: MAPPING CLINICAL OUTCOMES AND IMPLEMENTATION OF CLINICAL PRACTICE GUIDELINES

PLOS ONE

Dear Dr. Blanco-Mavillard,

Thank you for submitting your manuscript to PLOS ONE. After careful consideration, we feel that it has merit but does not fully meet PLOS ONE’s publication criteria as it currently stands. Therefore, we invite you to submit a revised version of the manuscript that addresses the points raised during the review process.

We look forward to receiving your revised manuscript.

Kind regards,

Itamar Ashkenazi

Academic Editor

PLOS ONE

Journal Requirements:

2. Please provide additional details regarding participant consent. In the ethics statement in the Methods and online submission information, please ensure that you have specified (1) whether consent was informed and (2) what type you obtained (for instance, written or verbal, and if verbal, how it was documented and witnessed). If the need for consent was waived by the ethics committee, please include this information.

3. Please refer to any post-hoc corrections to correct for multiple comparisons during your statistical analyses or sample size calculations performed. If these were not performed please justify the reasons. Please refer to our statistical reporting guidelines for assistance (https://journals.plos.org/plosone/s/submission-guidelines.#loc-statistical-reporting).

Reviewers' comments:

Reviewer's Responses to Questions

**Comments to the Author**

1. Is the manuscript technically sound, and do the data support the conclusions?

Reviewer #1: Yes

Reviewer #2: Partly

Reviewer #3: Yes

2. Has the statistical analysis been performed appropriately and rigorously? 

Reviewer #1: Yes

Reviewer #2: N/A

Reviewer #3: I Don't Know

3. Have the authors made all data underlying the findings in their manuscript fully available?

Reviewer #1: Yes

Reviewer #2: Yes

Reviewer #3: Yes

4. Is the manuscript presented in an intelligible fashion and written in standard English?

Reviewer #1: Yes

Reviewer #2: Yes

Reviewer #3: Yes

5. Review Comments to the Author

Reviewer #1: Dear author,

The article complies with the guidelines set by the magazine for its corresponding publication. The methodology is optimal. The conclusions are a true reflection of the objectives. It would qualify an update of the bibliographic articles over the last 5 years, especially due to recently published research. Still, congratulations.

Reviewer #2: Abstract

This is an observational study in three hospitals in Spain evaluation of all PIVC's inserted in adults admitted. This format is correct. The title and abstract are appropriate for the content of the text.

Introduction

Various articles are shown in the introduction but do not talk about the problem to be measured and it is necessary to delve deeper into the objective of the study. The bibliography of the articles referenced in the introduction is quite old. Currently, there are various studies and working groups in Spain that study the quality of intravenous catheters. It is necessary to update this bibliography.

Methods

The design is correct, and they collected data using a case report form that we had developed to analyze clinical outcomes for PIVC care based on the recommendation of the 2014 clinical practice guidelines.

It is not clear in the text that the same brand of intravenous catheters is used in the three hospitals; can be a limitation of the study.

Results

They observe significant differences between the characteristics of the context but do not specify whether they are in the same wards with the same pathologies between the three hospitals.

It highlights affirmations already known and does not bring any news.

Discussion and Conclusion

This is a study that brings together the technique of inserting intravenous catheters in order to achieve quality criteria based on clinical guidelines.

The hospital wards where the study was carried out are very diverse as well as the pathologies in which intravenous therapy had to be performed. Hospital policy was not taken into account regarding the quality of the technique or the information on adverse events.

The bibliography on which they are based is old and poor, and they leave aside important studies that are currently being carried out in Spain. They do not measure the totally adverse events of the technique and it does not contribute anything new.

Reviewer #3: This article adresses an relevant topic, especially when complications with PIVC can lead to unnecessary costs and prolong hospital stay. There are a number of issues with the methods and results that need to be clarified. Some of the conclusion does not supports the amount of data presented in the results. Below are more specific comments:

Abstract:

The conclusion that nearly 50% of patients did not know what a PIVC is, is interesting, but how this links with your reasearch question?

Introduction:

A bit more detail about the complications with PIVC and how clinical practice guidelines can improve care. The authors bring Health Literacy as a topic in the discussion, more information about it would provide welcome context here.

Methods:

• How did you choose these two guidelines? There was any criteria?

• How did you trained the external researches and for how long? How did you know that they were ready?

• Some technical details should be expanded and clarified to ensure that readers can understand exactly what the researchers studied.

Results:

• I do not see a link with the characteristics of the context and your research question.

• When you say that 76% does not present cognitive impairment but when you bring that nearly 50% does not know what a PIVC is, this 50% is about all your patients or about this 76%?

• Table 2. There is a lot of information that does not support your study.

Discussion:

• When you said that “The lack of context-related knowledge may be one of the most significant problems for implementation” you should discuss this better. Why it can be a problem?

• More details here would be welcome here about the clinical outcome indicators.

Conclusion:

• The conclusion does not support the results and does not fully answer your reasearch question.

6. PLOS authors have the option to publish the peer review history of their article (what does this mean?). If published, this will include your full peer review and any attached files.

Reviewer #1: **Yes: **Álvaro Astasio-Picado

Reviewer #2: No

Reviewer #3: No

---

## [Author Response · Author response to Decision Letter 0]

27 Aug 2020

Response to Reviewer’s and Editorial Comments

Reviewer reports:

Reviewer #1: 

Dear author,

The article complies with the guidelines set by the magazine for its corresponding publication. The methodology is optimal. The conclusions are a true reflection of the objectives. It would qualify an update of the bibliographic articles over the last 5 years, especially due to recently published research. Still, congratulations.

R: We are very grateful to the reviewer for their encouragement.

Reviewer #2: 

Abstract

This is an observational study in three hospitals in Spain evaluation of all PIVC's inserted in adults admitted. This format is correct. The title and abstract are appropriate for the content of the text.

Introduction

Various articles are shown in the introduction but do not talk about the problem to be measured and it is necessary to delve deeper into the objective of the study. The bibliography of the articles referenced in the introduction is quite old. Currently, there are various studies and working groups in Spain that study the quality of intravenous catheters. It is necessary to update this bibliography.

R: We appreciate the review, thank you. We have reflected upon this suggestion. We have left some old references due to its foundational relevance; however, we have also added new references (Guembe, 2018 and Saliva, 2018) underpinning our statement.

We are aware of the several active working groups developing this research line in our country that will sum up relevant evidence soon. The work by Guembe et al. is one in this line.

Methods

The design is correct, and they collected data using a case report form that we had developed to analyze clinical outcomes for PIVC care based on the recommendation of the 2014 clinical practice guidelines. It is not clear in the text that the same brand of intravenous catheters is used in the three hospitals; can be a limitation of the study.

R: Thank you for suggestion. We have now added the brand of the PIVCs and their devices in the methods section.

Results

They observe significant differences between the characteristics of the context but do not specify whether they are in the same wards with the same pathologies between the three hospitals. It highlights affirmations already known and does not bring any news.

R: As stated in the methods section, we selected all units in the three participating hospitals, describing the characteristics of the population of every hospital. Units were classified into “surgical” and “medical” depending on the profile. As we state, all clinical specialities except cardiac, thoracic, and neurosurgery are present in the sample of the three hospitals, being comparable in this respect. We also presented other relevant elements of the context, and it is in these elements where some differences have been found. We consider that these elements must be presented in a way that the reader can take them into account for the interpretation of the results. Our findings indicate that the clinical outcome indicators from CPG for PIVC care were moderate, highlighting differences between environments and types of hospitals. Our conclusions reiterate the crucial importance of PIVCs toward patient safety and experience and organisational quality, considering that this study is a baseline point for future studies with PIVCs in follow-up. 

Discussion and Conclusion

This is a study that brings together the technique of inserting intravenous catheters in order to achieve quality criteria based on clinical guidelines.

The hospital wards where the study was carried out are very diverse as well as the pathologies in which intravenous therapy had to be performed. Hospital policy was not taken into account regarding the quality of the technique or the information on adverse events.

The bibliography on which they are based is old and poor, and they leave aside important studies that are currently being carried out in Spain. They do not measure the totally adverse events of the technique and it does not contribute anything new.

R: We are grateful for this suggestion. Our study included in the method a section about PIVC care and maintenance. ‘Nurses inserted and maintained all PIVCs following the existing hospital policy, being like CPG recommendations. In summary, skin preparation before insertion was carried out with 2% chlorhexidine in 70% isopropyl alcohol. All PIVCs were Introcan SafetyTM (non-winged) catheters (B. Braun), with a needle-free valve directly connected to 10cm of extension tubing ending in a three-way connector (Becton Dickinson). A transparent dressing with polyurethane borders (TegadermTM, 3M) was applied at the insertion site to secure the PIVC in situ. Standard caps on all needleless connectors were in place to minimise accidental tubing disconnections’

Reviewer #3: This article addresses a relevant topic, especially when complications with PIVC can lead to unnecessary costs and prolong hospital stay. There are a number of issues with the methods and results that need to be clarified. Some of the conclusion does not supports the amount of data presented in the results. Below are more specific comments:

Abstract:

The conclusion that nearly 50% of patients did not know what a PIVC is, is interesting, but how this links with your research question?

R: Thank you for this suggestion, we have added the indicator and recommendation of the patient education in the table 1. The patient education on treatment targets, administration, infusion, associated complications and management of the catheter should be a crucial element of care of PIVC. Therefore, we considered relevant that our conclusion included this finding to highlight an under-studied clinical situation.

Introduction:

A bit more detail about the complications with PIVC and how clinical practice guidelines can improve care. The authors bring Health Literacy as a topic in the discussion, more information about it would provide welcome context here.

R: Thank you for suggestion. We have now added this paragraph ‘This gap is a complex and multifaceted phenomenon, which requires a deep understanding of decision-making[18]. The use of a knowledge mobilization model could counteract this situation, including strategies to promote fidelity to recommendations, audit and feedback of compliance and health literacy of vascular access, as crucial elements of a multimodal intervention [23]. Such gap is, therefore, a significant threat to patient safety and healthcare efficiency[24,25].’

Methods:

• How did you choose these two guidelines? There was any criteria?

• How did you trained the external researches and for how long? How did you know that they were ready?

• Some technical details should be expanded and clarified to ensure that readers can understand exactly what the researchers studied.

R: We are grateful for highlighting this relevant aspect. In October 2018, we published a systematic review of clinical practice guidelines (CPG) that provide recommendations for the care and prevention of adverse events associated with vascular catheter in adults. This study compared the quality of seven international CPGs, analysing methodological factors related to the process of CPGs development on the transference of knowledge. We selected the CPGs from the Spanish Health Ministry and the UK National Institute for Health and Care Excellence for their high-quality standards.

Concerning training of the external investigators, we have amended this point and now reads 'The external researchers received one-week of face-to-face training and a protocol for completing the case report form. Also, they completed a full working day evaluation with a mentor before starting the study.

Results:

• I do not see a link with the characteristics of the context and your research question.

• Table 2. There is a lot of information that does not support your study.

R: Thank you for suggestion. The purpose of this study was to analyse the clinical outcome indicators from CPG for the insertion, maintenance, and management of PIVCs on different hospital types and environments. Therefore, we considered that it was necessary to describe the characteristics of the context as key elements for implementation and for the interpretation of the study results.

• When you say that 76% does not present cognitive impairment but when you bring that nearly 50% does not know what a PIVC is, this 50% is about all your patients or about this 76%?

These findings referred to patients who did not have cognitive impairment. The text reads 'In our sample, 474 patients (76%) did not present cognitive impairment. Among these, 250 patients (52.7%) recognised and identified the inserted PIVC'.

Discussion:

• When you said that “The lack of context-related knowledge may be one of the most significant problems for implementation” you should discuss this better. Why it can be a problem?

• More details here would be welcome here about the clinical outcome indicators. 

R: We appreciate the recommendation, thank you. The discussion has been revised, and now reads ‘The lack of context-related knowledge may be one of the most significant problems for implementation strategies into the healthcare system, being itself a fundamental limitation that questions the efforts made to improve the quality of interventions and the fidelity on the use of evidence-based practice[50]. The adaptation of interventions to local contexts is an indispensable element in the science of implementation. However, it is frequently not explicitly considered how local context factors determine its success[49]. In-depth knowledge of nurse participation in hospital affairs, leadership, size of nursing teams and professional relationship, added to the use, attitudes, and knowledge of professionals towards evidence-based practice will provide relevant information about the contextual and individual mechanisms[51,52] that can facilitate the integration of tacit and explicit knowledge into decision-making improving adherence to best available recommendations[49,53]’.

Conclusion:

• The conclusion does not support the results and does not fully answer your research question.

R: Thank you for suggestion. We considered that the findings answer adequacy our research question. The conclusion reads ‘Our findings indicate that the clinical outcome indicators from CPG for the insertion, management, and maintenance associated with PIVC were moderate, highlighting differences between hospital environments and types. Also, we observed that almost 50% of patients did not know what a PIVC is. These findings reflect a wide gap between knowledge and optimal clinical practice, which would explain the moderate adherence to PIVC care and the need to use a knowledge transfer model with contextual mechanisms and individual factors.’

---

## [Decision Letter · Decision Letter 1]

21 Sep 2020

CARE OF PERIPHERAL INTRAVENOUS CATHETERS IN THREE HOSPITALS IN SPAIN: MAPPING CLINICAL OUTCOMES AND IMPLEMENTATION OF CLINICAL PRACTICE GUIDELINES

PONE-D-20-15991R1

Dear Dr. Blanco-Mavillard,

We’re pleased to inform you that your manuscript has been judged scientifically suitable for publication and will be formally accepted for publication once it meets all outstanding technical requirements.

Kind regards,

Itamar Ashkenazi

Academic Editor

PLOS ONE

Reviewers' comments:

Reviewer's Responses to Questions

**Comments to the Author**

1. If the authors have adequately addressed your comments raised in a previous round of review and you feel that this manuscript is now acceptable for publication, you may indicate that here to bypass the “Comments to the Author” section, enter your conflict of interest statement in the “Confidential to Editor” section, and submit your "Accept" recommendation.

Reviewer #2: All comments have been addressed

2. Is the manuscript technically sound, and do the data support the conclusions?

Reviewer #2: Yes

3. Has the statistical analysis been performed appropriately and rigorously? 

Reviewer #2: Yes

4. Have the authors made all data underlying the findings in their manuscript fully available?

Reviewer #2: Yes

5. Is the manuscript presented in an intelligible fashion and written in standard English?

Reviewer #2: Yes

6. Review Comments to the Author

Reviewer #2: (No Response)

7. PLOS authors have the option to publish the peer review history of their article (what does this mean?). If published, this will include your full peer review and any attached files.

Reviewer #2: No

---

## [Editor Report · Acceptance letter]

25 Sep 2020

PONE-D-20-15991R1 

Care of peripheral intravenous catheters in three hospitals in Spain: mapping clinical outcomes and implementation of clinical practice guidelines 

Dear Dr. Blanco-Mavillard:

I'm pleased to inform you that your manuscript has been deemed suitable for publication in PLOS ONE. Congratulations! Your manuscript is now with our production department. 

Kind regards, 

on behalf of

Dr. Itamar Ashkenazi 

Academic Editor

PLOS ONE